# Representations of Sexuality among Persons with Intellectual Disability, as Perceived by Professionals in Specialized Institutions: A Systematic Review

**DOI:** 10.3390/ijerph19084771

**Published:** 2022-04-14

**Authors:** Tamara Guenoun, Barbara Smaniotto, Christophe Clesse, Marion Mauran-Mignorat, Estelle Veyron-Lacroix, Albert Ciccone, Aziz Essadek

**Affiliations:** 1Centre de Recherche en Psychopathologie et Psychologie Clinique (CRPPC), Université Lumière-Lyon 2, 69365 Lyon, France; smaniotto.barbara@yahoo.fr (B.S.); marion.mauran-mignorat@univ-lyon2.fr (M.M.-M.); e.veyronlacroix.psycho@gmail.com (E.V.-L.); a.ciccone.99@gmail.com (A.C.); 2Center for Psychiatry, Wolfson Institute of Preventive Medicine, Barts & The London School of Medicine & Dentistry, Queen Mary University of London, London E1 4NS, UK; christophe.clesse@hotmail.fr; 3Laboratoire Inter-Universitaire de Psychologie (LIP), Université de Grenoble-Alpes, 38058 Grenoble, France; 4Laboratoire INTERPSY, Université de Lorraine, 54052 Nancy, France; aziz.essadek@univ-lorraine.fr

**Keywords:** systematic review, intellectual disability, sexuality, mental representations, professionals, specialized institution

## Abstract

The objective of this systematic review is to make an inventory of the representations of the professionals of specialized institutions on the sexuality of persons with intellectual disabilities. The scientific studies were identified according to the PRISMA protocol using 18 databases, with keywords on sexuality and parenthood. Studies were reviewed through a methodological assessment and then a thematic analysis. Twenty-four studies were reviewed and three themes were identified: professionals’ representations of gender, sexuality, and consent; professionals’ perceptions of their role in supporting people’s sexual lives; and the ways in which professionals construct representations of people’s sexual lives. This corpus highlights deep paradoxes in the representations of professionals concerning the socio-affective needs and sexuality of people with intellectual disabilities, creating what we could define as a “system of incompatibility” and leading to difficulties in positioning. Support is still too random and subject to control logics in the name of protecting users, who are perceived as vulnerable. Training and new ways of teamwork appear to be central to supporting the evolution of the representations and practices of professionals. Future research anchored in practices and involving users as well as professionals is necessary to better understand the paradoxical aspects of professionals’ representations and to draw alternative ways of constructing these representations.

## 1. Introduction

While the right to sexuality and an emotional life for individuals with intellectual disability is now widely recognized by global bodies such as the United Nations (*The standard rules on the equalization of opportunities for persons with disabilities*, 1994; United Nations, 2008) [1,2], its implementation remains seriously hampered. Many international studies—as well as a few studies undertaken in France—have focused on the measures that support the emotional and sex lives of such people and on the training that the professionals who care for them require. The international studies undertaken on the issue have given rise to a vast body of literature [3], ranging from instrumental approaches to sexuality to reflections that focus more on the love lives of people with an intellectual disability. These individuals have therefore been the subject of numerous studies on the potential obstacles to their sex lives and with regard to their social and emotional needs. However, practices on the ground have been slow to evolve [4]. Indeed, while researchers and practitioners have focused on the creation of programs that both professionals and these individuals themselves can use, there seems to be a paradox here [5,6]. This paradox is between professionals’ recognition of good practices concerning one’s sex and emotional life, which one can choose freely, and the functioning of the institutions that care for people with disabilities, which occasionally deny them their rights. In France, several researchers [7,8] have highlighted the existence of several preconceptions among professionals working in specialized institutions, including educators, carers, nurses, and psychologists. The sexuality of individuals with intellectual disabilities is still perceived negatively and in a very binary way. Indeed, it is seen either as non-existent among individuals viewed as asexual, or as bestial—a mere instinctive impulse [9,10,11].

This systematic review seeks to analyze the representations of professionals from specialized institutions (educators, carers, nurses, psychologists, executives, directors of structures) with regard to the sexuality of individuals with intellectual disabilities, and how these representations affect their perception of these individuals’ emotional lives. A better understanding of these representations may shed new light on how the sex lives of individuals with an intellectual disability (ID) may be supported, and thus promote the evolution of practices. How do these professionals perceive the sexuality of the people they support on a daily basis, and how do they perceive their role, especially with regard to the emotional and sex life projects of these individuals?

## 2. Method

### 2.1. Research Question

This systematic review is part of the “Intellectual Disability and Parenthood” study funded by INSERM (1811034-00). The objective of this project is to gain a better understanding of the specific needs and desires—in terms of the emotional, sexual, and parental lives of people with intellectual disabilities living in medical and social centers—in order to propose alternative support options for residents and innovative training for professionals.

This research project was motivated by clinical questions arising from field observations: residents of medical and social centers hardly expressed their desire for an emotional life; even less did they express the desire to have children. These findings are inconsistent with the results in the general population, where these desires are considered to be very common and their absence atypical [12]. Why is it that adults in these centers hardly talk about these aspects of their intimate lives?

Given the dynamic aspect of this research, which addresses the themes of parenthood, sex lives, representations, professional practices, and ethical aspects, we chose to undertake a broad systematic review regarding these different dimensions. The results presented here fall within a thematic subsection; indeed, the international articles published on these themes are quite abundant. This paper will therefore examine the representations of professionals concerning the sex lives of those people with ID with whom they work.

### 2.2. Systematic Review Protocol

The methodological protocol of this study is part of the DEFIPARENT project (Alternative support measures for adults/Intellectual Disability and Parentality) funded by INSERM (1811034-00). The objective of this project is to gain a better understanding of the specific emotional, sexual, and parental needs and desires of people with ID living in medical and social centers. This will provide new reflections on the training of professionals and help meet residents’ needs through the building of new infrastructure that is more adapted to their life projects.

To this end, the authors first outlined a general scientific overview, based on the PRISMA protocol [13] and registered under the reference number PROSPERO CRD42020127294.

The following list of keywords in French and English—“Mental disorder”, “intellectual disability”, “learning disability”, “mental disability”, “psychical disability”, “mental retardation”, “deficiency”, “handicap”, “handicap psychique”, “handicap mental”, “déficience intellectuelle”, “polyhandicap”—was associated with this second list of keywords: “parenting”, “parenthood”, “parentality”, “parent”, “pregnancy”, “procreation”, “sexuality”, “reproductive health”, “childbirth”, “contraception”, “child desire”, “intimacy”, “intimity”, “family”, “parentalité”, “parent”, “procréation”, “contraception”, “sexualité”, “grossesse”, “naissance”, “désir d’enfant”, “intime”, “intimité”, “famille”.

After removing the duplicate words (for example, “parent” in French and “parent” in English), a total of 55 French and 98 English combinations were retained. These combinations were then analyzed again in 18 databases (Figure 1).

Each phase of the systematic review process (identification, selection, eligibility, and inclusion) was updated every other month. For each phase, all possible discrepancies were assessed and the entire team participated in joint decision making. A total of 5,009,747 references were screened during the identification phase. Of these, 2698 titles were selected, a number that dropped to 2058 when duplicates were eliminated on the basis of the scope and the exclusion criteria mentioned above. Across all databases, the minimum percentage of similarity in the selection of titles (the same title selected by both examiners) was 92.3%.

We agreed upon several inclusion and exclusion criteria. Firstly, the year of publication is not considered as an exclusion criterion. To understand the possible evolution of professional representations concerning the sexuality of people with ID, no specific publication period was selected. Secondly, all of the books, book chapters, commentaries, literature reviews, conference abstracts, letters to the editor, web pages, and articles where these words were lacking from the abstracts were excluded. Thirdly, during the selection phase (based on a reading of the abstracts), the research team excluded articles out of the scope. We thus excluded research focusing on mental disorders (for example, schizophrenia, bipolarity, borderline personality disorder) that were not associated with ID. ID involves impairments of general mental abilities [14] that impact adaptive functioning in three domains: the conceptual domain, the social domain, and the practical domain. It is diagnosed based on the severity of deficits in adaptive functioning [15]. The disorder is considered chronic and often co-occurs with other mental conditions such as attention-deficit/hyperactivity disorder, learning disabilities, and autism spectrum disorder. For that reason of co-occurrence, we included articles focusing on these mental conditions and checked during the inclusion phase of the systematic review whether the population studied in these studies had a co-occurrence of ID. In particular, several articles on learning disabilities were included in this systematic review. This decision was motivated by the fact that people with these disabilities are often present among people with ID in specialized institutions. Learning disabilities result from impairments in one or more processes related to perceiving, thinking, remembering, or learning. These include, but are not limited to, language processing; phonological processing; visual spatial processing; processing speed; memory and attention; and executive functions (e.g., planning and decision making [15].

Articles focusing on parents (without disabilities) of children with ID were also excluded. As a result, 897 articles were included in the eligibility phase of our research. Finally, during the fourth and last phase of the systematic review, we excluded quantitative studies and focused on qualitative research. Our goal in this systematic review was to allow a better understanding of the subjective experience of professionals. We believe that this is a central aspect of a rather complex issue.

Given the high number of publications selected, we added an additional phase of thematic categorization [16]. Six thematic categories were established (Figure 1): Parenthood; Sexuality; Evaluation of Tools; Training; Ethics and Law; and a Medical and Legal dimension. Several articles fell within more than one category and were therefore classified into several themes. This article focuses on the “Sexuality and intellectual disability” category, which comprises 484 articles. In order to address more specific aspects, we drew on the PRISMA systematic analysis method. From these 484 references, we extracted those dealing with the representations of healthcare professionals, and with support and education offered regarding the sexuality of people with ID under their care. This article is therefore part of a series of upcoming articles. It focuses specifically on the section “Representations of the sexuality of people with intellectual disability among the professionals who care for them”.

### 2.3. Study Selection

At the end of the second phase of the systematic review, 96 articles corresponded to the theme of representations of the sexuality of people with ID among the professionals who care for them. The articles were therefore read in depth. At the end of this detailed reading, 72 articles were excluded.

A total of 40 were out of scope:-Some of these focused on professionals working with mentally ill persons with no co-occurrence of ID.-Other studies focused on the representations of the families of individuals with ID, on these people’s representations of contraception, on students’ representations, or on representations on the subject of the general population.-Several articles did not directly address the issue of sexuality, but rather focused on related themes: parenthood, general ethical issues, or an analysis of scholarly writings.-Others focused on the sexuality of adolescents with intellectual disability.

We also excluded 10 studies because of their methodological weakness, including an item eliminated that was a duplicate that had initially gone unnoticed. We used the CASP tool [17] to evaluate the methodology of the studies. Mostly, the excluded studies had a poor research design, with no clear data collection. Moreover, the data analysis was not rigorous enough.

Finally, we excluded 22 quantitative studies, using surveys and questionnaires from qualitative ones based on research interviews and analyzed using different methodologies.

We thus identified 22 qualitative studies, based on research interviews analyzed using different methodologies, and 2 studies using a mixed methods research design, in which a rating scale analysis was used to analyze the research interviews.

This systematic review therefore includes 24 articles based on qualitative methodology, including 2 based on a mixed methods research design, focusing on the representations of professionals in medical and social centers regarding the sex lives of adults with ID (Table 1).

## 3. Results of the Systematic Review

### 3.1. Overview of the Corpus

The vast majority of these articles were from English-speaking countries (Canada, USA, Australia). One study was undertaken in France, one in Nigeria, one in Poland, and one in South Africa.

Most of the professionals interviewed in the articles were psychologists, educators, teachers, social workers, care home managers, and nurses. One study focused specifically on medical staff. Two studies focused on education professionals.

The majority of these studies involved research interviews. Two studies used observation alongside the interviews, thus adopting an ethnographic approach. Three set up parallel research groups involving professionals.

In terms of the tools used to analyze these interviews, observations, and groups, five studies used interpretative phenomenological analysis, two employed grounded theory methodology, one was a case study, one used an ethnographic approach, and 13 undertook discourse and thematic content analysis.

### 3.2. Thematic Analysis

The analysis of the data presented in these different articles reveals the existence of several themes:-Professionals’ representations of gender, sexuality, and consent;-Professionals’ perceptions of their role in supporting peoples’ sex lives;-The construction of professionals’ representations of individuals’ sex lives.

#### 3.2.1. Professionals’ Representations of Gender, Sexuality, and Consent

Overall, professionals perceive people with ID as a very high-risk group with regard to sexuality issues, and particularly vulnerable to sexual violence. The representations appear to be highly gendered [40] and very stereotypical of the behaviors expected, and perceived as acceptable, in men and women. Thus, professionals’ fear of rape and sexual violence is focused on women [19]. Individuals are categorized into two groups: one with an “angel-like” sexuality, characterized by asexuality, the other with a “beast-like” sexuality, meaning impulsive and bestial [19]. In the latter case, and in view of this distinction, the sexuality of people with ID may be judged negatively [19,40]. More than 20 years ago, professionals tended to consider that people had greater knowledge and experience of sexuality than they actually did [41]. Is this still true today, given sexual liberation and greater access to online sexual content?

Beyond the aspects of sexual life, the majority of professionals have long been concerned with the emotional lives of people with regard to issues of marriage and parenthood [21]. Contraception is therefore a central issue in regulating access to parenthood for people with ID. However, decades ago, this contraception [35] was given by healthcare professionals through standardized care protocols. For instance, women were forced to take the contraceptive pill irrespective of their life project. Roy et al. [35] show that the adaptation of contraceptive methods to an individual’s life plans by listening to the feedback from their family circle may have a positive impact. Such an approach means that professionals may be better able to understand the needs of women with ID and therefore to propose adapted contraceptives that are more willingly accepted. Further research will help to shed light on current practices in this regard.

#### 3.2.2. Professionals’ Perceptions of Their Role in Supporting People’s Sex Lives

Professionals have a rather contradictory perception of their role. Indeed, they feel that it is their duty to facilitate the sex lives of people under their care, but at the same time to protect them from the potential dangers of this sexuality [29,30]. In this regard, they struggle with the concept of consent [29]. They experience their positions in specialized institutions as somewhat peculiar because they feel that they have to decide whether or not to allow emotional and/or sexual relationships between residents. Moreover, several articles have highlighted the need to develop a new standard of competence among professionals, capable of taking into account “individuals’ capacity of consent” [29,38].

Professionals also appear to find it difficult to include these people in making decisions about their own sex lives and to let them freely choose their relationships with others, especially where homosexuality is concerned [39]. They also exhibit a fear of pregnancy [38].

Generally speaking, the articles analyzed highlight a lack of communication and dialogue on sexuality between professionals and people with ID [36]. The “incident culture” dominates [18,37]: irrespective of their department, professionals only address sexual issues when there is a problem [29].

It is also worth mentioning that the specific needs of women, and of people with moderate to severe ID, are neither addressed nor taken into account, unless as a last option [18,37].

Several studies note great tension between incident management and individuals’ right to sexuality [29,34,39]. This tension creates contradictory feelings among professionals [33], to the extent that supporting those in their care in relation to these issues is described as a “clever juggling act” [32]. Faced with these conflicting situations, professionals are thought to keep the issue of sexuality at bay [32], or even to function as disciplinary “new institutional walls” [31]. They therefore adopt a position of power in the name of protection, which leads those under their care to react in two similarly contradictory ways: either by accepting or rebelling against the professionals [27]. Indeed, people with ID say that they feel as if they are closely monitored and constantly have to ask for permission [25,34]. While these individuals appear to have internalized a discourse that limits their choices, the study by Bates et al. [20] points out that these implicit interdictions have changed in recent years, and now lean toward more subtle forms of pressure regarding choosing one’s sex and emotional life [20]. Further studies on how professionals restrain these individuals seem necessary.

In short, these studies report a tendency to control, rather than to welcome and support, the choices of these individuals around their emotional and sex lives [27]. It therefore appears that professionals need training on how to support the emotional and sex lives of people with ID [25,36].

#### 3.2.3. The Construction of Professionals’ Representations of People’s Sex Lives

Given that there is no “guide” to understanding the issue of the sexuality of the people under their care, the representations of professionals appear to have been constructed through influence: based on the institution’s culture, the dynamics of the professional group, the social context, or according to one’s own experience and personal values [22,32]. The influence of the social environment is particularly emphasized [23]. Professionals are strongly influenced by the social, cultural, and religious framework in which they operate. They tend to conform to this framework, at the risk of finding themselves out of step with their initial representations of proposing free choice to the people under their care in terms of their emotional and sex lives [23]. In this respect, researchers may become the bearers of this social pressure. Professionals’ responses concerning their support for the emotional and sexual lives of people with ID appear to vary widely depending on the positions that researchers adopt [23]. Specifically, Brown [22] outlines four types of belief systems among professionals and peer helpers: advocates, supporters, regulators, and humanists. How, then, are the beliefs of each caregiver translated in a professional context? This question would require further research.

It appears essential to raise awareness and to train professionals in the specific needs of this public [30]; it also seems important to create a positive space for people with ID with regard to sexuality [25,36]. This seems even more important for this new generation, which has grown up with more sexual freedom and the idea that they have the right to choose what they do with their bodies [31].

First, it should be noted that professionals’ training may have a moderate impact in cases where:-Not all professionals in a team are trained;-The management does not reorganize the center to allow a genuine application of what is learned in training (rooms for couples, intimate spaces to allow people to meet and to have sexual relationships, etc.);-The social–cultural context still stigmatizes sexuality and disability, especially because of the fear of sexual assault [28].

Second, the training programs presented in the articles analyzed here have come under criticism: they are thought to lack substance [38] and to be difficult to put into practice [24]. Indeed, these programs are caught up in several ideologies, each with their own shortcomings:-Discourse of protection vs. discourse of normalization;-Biologizing discourses on sexuality, where sex is differentiated from intimacy and pleasure, discourses that contrast with the needs expressed by people seeking to encounter the other;-Professional discourses in terms of competences, even though few competences are clearly identifiable and explicable in this field. Social–emotional needs and sexual desire are, by their very nature, complex. They primarily involve relational skills, i.e., collaborating with people, and are determined by the professionals’ intuitions and ways of being [24].

In view of these elements, it seems necessary to design collaborative training with professionals based on the testimony of the users themselves [24]. Indeed, taking into account the discourses and needs of professionals may lead to a change in the dominant discourses in specialized centers, by freeing professionals from their isolation through activities allowing them to work more closely with families [26,29].

## 4. Discussion

This systematic review reveals that professionals’ representations on the subject appear to be strongly marked by several generally negative viewpoints regarding the sexuality of people with ID:-A perception of the sexuality of people struggling with gender stereotypes;-Difficulty thinking positively about the aspects of people’s lives as a couple or as a family;-The feeling that they must protect individuals under their care from great violence.

These aspects are directly associated with a fear of incidents and with risk prevention. The construction of these viewpoints is based primarily on the influence between colleagues, in line with moral or religious values.

However, all of these studies also shed light on how, when they come into contact with those under their care, professionals are committed to respecting these people’s rights to an emotional and sex life.

There is, therefore, a large gap among professionals between representations that portray a desire for action (aimed at supporting the informed choice of people on this issue) and the fact that they are limited in their actions by an institutional, social, cultural, and religious framework that bears a negative and inhibiting vision [23]. Different representations of the families of people with ID may be an additional limitation [23]. Professionals feel like they are undertaking a “clever juggling act” [32] when it comes to supporting the emotional and sex lives of people with ID: they must make complex and sometimes difficult choices between often contradictory representations, built on discourses and practices that are themselves contradictory.

Quantitative studies corroborate these findings. These studies highlight the fact that professionals have rather positive intentions regarding the sex and emotional lives of the people with ID under their care, and that these professionals are influenced by their age, religion, and level of education [42,43,44,45,46,47,48]. However, these increasingly positive mental intentions and representations are hampered by institutional dynamics that tend to be limiting and also by the considerable need for training [49,50,51,52,53]. Most of the studies on this theme highlight the need for the continuous training of professionals on these issues.

The entire corpus highlights a profound paradox between the representations of professionals concerning the social–emotional needs and sexuality of people with ID, and the possible practices in the field. Most of the articles underline the importance of training professionals on these sensitive issues. However, this systematic review shows that the professional training suffers from the same paradoxical approach. The training tends to focus on one or two aspects, too narrow to help the professionals to face the complex situations they have to deal with in practice. This makes it difficult for professionals in the fields of health, social education, and care to help individuals with ID to make informed choices about their sex lives and about their social and emotional needs.

We decided against limiting our analysis to a specific timeframe in order to examine the changes in societal mores as well as contemporary changes. While society has changed and now pays greater attention to the emotional and sexual lives of vulnerable people, professionals’ questions and difficulties in positioning themselves remain major, and still valid, concerns. Finding the right balance in supporting the emotional and sex lives of people with ID is still a very complex question when it comes to the relationship between help, care, and education.

Two opposing viewpoints thus emerge, and these appear to be constant through time: a societal and institutional context geared toward the absolute protection of individuals, and representations more rooted in the empowerment of people with regard to these intimate aspects of their lives. From a practical point of view, the difficulty in dealing with this paradox leads professionals to a position that seems impossible to disentangle: recognizing human rights, but denying the people under their care their right to a sex life through control-based practices. In this context, the institutional framework does not appear to take on a regulatory role: it is up to professionals to come to terms with—or, rather, to manage—these paradoxical representations. As a result, it is the team, i.e., colleagues, who act as a frame of reference to reflect on these representations, through strategies of influence, accentuated by the disparity of training and values specific to each professional.

These findings are consistent with the results of studies on teamwork in medical and social institutions [54]. These dichotomies and paradoxes in the support of the emotional and sex lives of the people cared for relate to the notion of a “system of incompatibility”. This notion is defined as the consciousness of the presence of two fields that are unable to unite and connect with each other, even though they concern the same category of people. It may thus be assumed that there is a “system of incompatibility” between the protection of vulnerable people on the one hand, and the facilitation of their emotional and sex lives on the other. Fustier [54] suggests that this system of incompatibility is the result of an unconscious script within the teams, rooted in the culture of the institution and the history of its origins. It can give rise to the organization of certain teams, structured around divisions or denials [55], which can lead to some form of violence in the support proposed [56]. In this regard, the literature highlights the extent to which the issue of people’s sex lives is shrugged off [18,36,37] in order to avoid the embarrassment it arouses. In this sense, what the institution and professionals perceive as an “incident” may ultimately be the sign of an extreme, desperate form of expression of a sexuality, of a desire hitherto controlled, frustrated, or ignored. However, it seems necessary to distinguish between the denial inherent in the institutional organization of these centers and that associated with the responsibility of the professional team, i.e., the strategies they adopt in the face of the multiple puzzles they encounter in providing daily support for the emotional and sex lives of people with ID.

Lastly, the paradoxes that cut across professionals’ representations of people with disabilities may help to shed light on their lack of support for, and preparation of, the people under their care [30,57]. While the latter need to be able to rely on identifying models to construct themselves [58], they still suffer today from an absence of reference points to help them to address (and especially to live) their emotional and sex lives, to make enlightened decisions, and to empower themselves. There is a clear need for emotional and sex life education for people with ID [43]. This would require careful consideration of these two complementary dimensions from the outset by the professionals who support them. Two decades ago, it was shown that professionals have the greatest impact and influence on people with ID during those years when their sexuality is developing [59].

## 5. Conclusions

Professionals’ difficulty in positioning themselves in the face of the emotional and sex lives of people with ID means that the support they propose is still too uncertain and leans toward control in the name of protecting users who are fundamentally perceived as vulnerable. The global training of professionals and management teams appears to be an effective path to supporting the evolution of professionals’ representations (and practices) regarding the sex and emotional lives of people with ID. However, this systematic review has shown that it is not easy for training programs to develop and teach good practices on such a complex subject. This review opens up new perspectives for research and reflection on training methods that take into account the paradoxical nature of the representations that influence support for the emotional and sex lives of people with ID. It seems important to promote studies rooted in practices and involving users and professionals.

## Figures and Tables

**Figure 1 ijerph-19-04771-f001:**
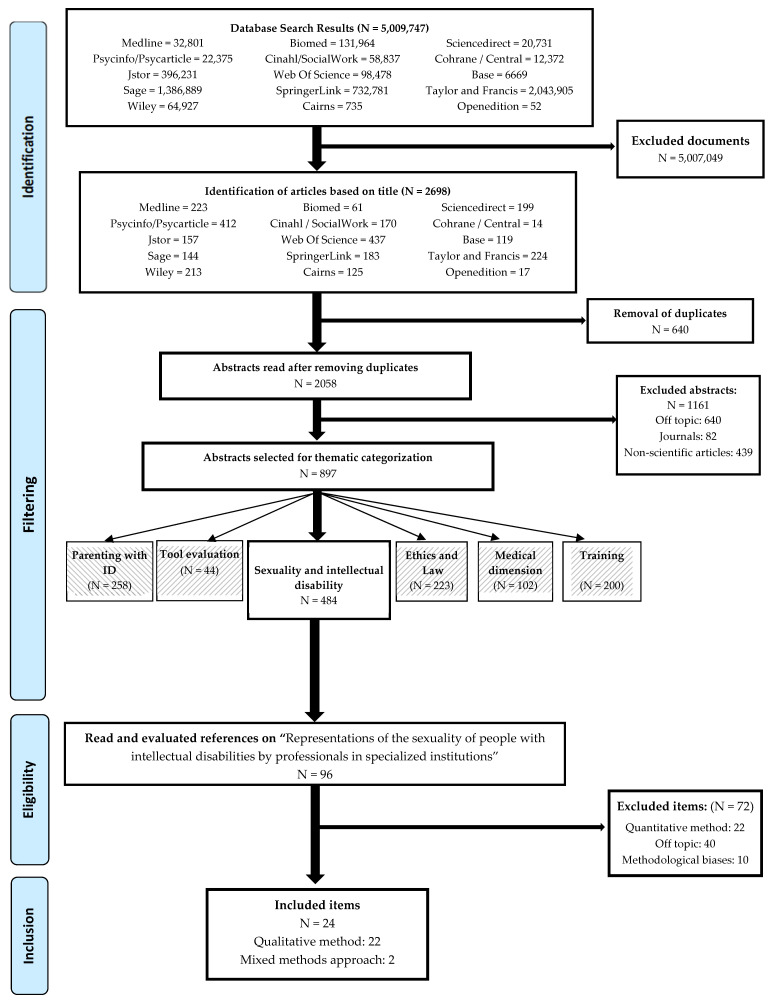
Flow diagram of the selection process relating to representations of the sexuality of people with intellectual disabilities by professionals in specialized institutions.

**Table 1 ijerph-19-04771-t001:** Articles included in the systematic review on “representations of the sexuality of people with intellectual disabilities by professionals in specialized institutions”.

	Study Citation and Country	Aim	Readership	Method of Data Collection	Key Findings
1	[18] Thompson, V.R. et al. Australia	To examine the barriers to sexual health provision of people with ID as experienced by disability service providers and clinicians.	Eight managers (four men, four women) and 23 clinicians (8 men, 15 women).	Semi-structured interviews carried out in two phases, with managers of services receiving people with disabilities by telephone, then with professionals in the field face to face. Both are analyzed with constructivist grounded theory.	The difficulties identified in the research have been grouped into three broad categories: administration (lack of funding and lack of policy guidelines), attitudes (myths about the sexual health) and experiences (lack of staff training).
2	[19] Aderemi, T.J. Nigeria	This article reports on teachers’ views on the sexuality of Nigerian learners with intellectual disabilities and their awareness of their risk of HIV infection.	Twelve specialized teachers (nine women and three men) working with people with ID. The age of the people with ID with whom they work is not specified.	Semi-structured interviews with 12 specialized teachers (9 women and 3 men) working with people with ID. Analyzed with interpretative phenomenological analysis.	Teachers feel able to teach about sexuality, they perceive sexuality as negative. They are concerned about the risk of rape of women with ID, and they have the belief, shared by families, that a child from a woman with ID would help the family recover from the disappointment of having a child with ID.
3	[20] Bates, C., et al. United Kingdom	To understand some of the barriers that people with learning disabilities face in relationships and to consider what changes professionals could make to address these.	Eleven people, aged over 18, identified as having an ID.	Case studies illustrating a number of themes relating to the support that people with learning disabilities received and needed from staff to develop and maintain relationships.	People with learning disabilities continue to experience barriers with regards to relationships. Their rights and choices are not always respected and a climate of risk aversion persists in areas such as sexual relationships. The research highlighted the balancing act staff must engage in to ensure that they remain supportive without being controlling or overprotective of individuals in relationships.
4	[21] Brantlinger, E. United States	To explore the attitudes of people in various professional fields toward sexuality, sterilization, and people with disabilities.	Legal professional *N* = 14 Medical professional *N* = 8 Professional of family planning agency *N* = 4 Professional of department of Public Welfare *N* = 3 Professional of developmental disabilities agency *N* = 13 Educators *N* = 8	Semi-structured interviews were used to explore the attitudes of legal, medical, social welfare, development agency and education professionals towards sterilization of people with disabilities.	Most professionals had been involved in some capacity with sterilization cases of people with disabilities and were concerned about present practices.
5	[22] Brown, R.D.; Pirtle, T. United States	To describe the perceptions of involved adults concerning the sexuality of individuals with intellectual disabilities.	Forty individuals who provide direct care or instruction to individuals with intellectual disabilities completed the 36-item Q-sort.	The Q-methodology was chosen. This method combines qualitative strategies with quantitative and qualitative analysis.	An understanding of their feelings allows caregivers and educators to react to individuals under their care in a professional manner, even when the situation may challenge their personal beliefs. For the working professional, communication could also be enhanced with an understanding of commonly held beliefs and opinions. The understanding of others’ beliefs could provide a frame of reference under which communication can proceed with parents, teachers, and the individual with intellectual disabilities.
6	[23] Caresmel, N. France	To describe how professionals in the medico-social sector see the sexuality of people with intellectual disabilities.	39 professionals working in institutions for people with intellectual disabilities.	Semi-structured interviews were used with professionals. Social representation theory was used to analyze the data.	The results show a strong discrepancy between a representation conveying a desire for action, a recognition of the reality of manifestations, demands and limits associated in particular with the absence of an explicit institutional framework, and different family representations.
7	[24] Chivers, J.; Mathieson, S. Australia	This article examines some of the dominant discourses that impacted on the process of developing a curriculum for staff working with people with an ID.	Guides to publishing a training course by two professionals in the field.	Analysis of dominant discourses on sexuality in professional training programs.	Constructions of sexuality as being solely a biological function, sex as dangerous, and sex as penetration are challenged, as are some of the dominant discourses of learning.
8	[25] Ćwirynkało, K. et al. Poland	To know how professionals working with adults with ID view the sexuality and intimate relationships of adults with ID.	Interviews with 16 professionals from several day and residential centers.	The authors conducted 16 interviews with professionals from several day and residential centers. The data were analyzed using the phenomenographic method.	First, professionals shape the quality of life of the clients where they work. The attitudes of caregivers and support workers directly influence people with ID. Second, the attitudes of professionals definitely affect the relationships of those professionals with their clients’ parents or caregivers. Last but not least, the existence of negative attitudes and misconceptions about the sexuality of individuals with ID among professionals can lead those individuals to internalize such attitudes and misconceptions.
9	[26] Eastgate, Gillian; Scheermeyer, Elly; van Driel, Mieke L.; Lennox, Nick Autralia	This study sought information from people involved in the care of adults with ID regarding how they supported them in the areas of sexuality, relationships and abuse prevention.	Twenty-eight family members and paid support workers caring for adults with intellectual disabilities.	Semi-structured interviews and focus groups were held with 28 family members and paid support workers caring for adults with intellectual disabilities.	Major themes emerging included views on sexuality and ID, consent and legal issues, relationships, sexual knowledge and education, disempowerment, exploitation and abuse, sexual health, and parenting.
10	[27] Grace, N. et al. United Kingdom	This study explores representations of staff discourses about the sexuality and intimate relationships of patients with intellectual disabilities.	Interviews were conducted with eight individuals with ID.	Semi-structured interviews were carried out with individuals with ID and analyzed using the principles of critical discourse analysis.	Discourses around sex appear to serve the interests of staff and the hospital, while being restrictive and often incomprehensible to service users. Implications for service development, and future research directions, are considered in the context of “Transforming Care”.
11	[28] Hanass-Hancock, J. et al. South Africa	The paper discusses the educators’ understanding and experiences of using an innovative sexuality training approach for educators of learners with diverse disabilities (Breaking the Silence).	A total of 13 educators (including 9 teachers, 2 psychologists, and 2 principals) were provided with the training and resources of the Breaking the Silence approach.	Following a 12-month implementation period, in-depth interviews were conducted with the 13 educators.	Although educators were able to implement parts of the approach, contextual factors impacted the degree of implementation. These factors were related to perceptions of socio-cultural norms, interpersonal engagement with peers and management, the structural environment of school settings, and the wider community setting. Educators began to address cultural taboos related to talking about sexuality, but were challenged by untrained staff and the larger socio-cultural context, which includes a heighted risk of sexual violence against their learners.
12	[29] Keaser, F., United States	This article examines, in detail, all aspects of consent and ways for determining consent as well as the responsibilities an interdisciplinary team has for managing mutual sex behaviors.	Team discussions generated by the situation of two men who want to have a sexual relationship in a care home.	Case study.	It advocates for a new standard of competence in sexual relations between two severely mentally retarded persons to be established.
13	[30] Lafferty, A. et al. Ireland	To understand how the barriers of the attitudes and perceptions of family carers, frontline support workers and professional staff toward the participation of persons with intellectual disabilities in relationships and sexuality education (RSE) might be reduced.	Study included 22 carers, 24 professionals from various disciplines working in the field of intellectual disabilities: 8 learning disability nurses, 5 social workers, but including also educationalists, social care managers and other healthcare professionals such as a psychiatrist, a psychologist and an occupational therapist; 24 frontline staff recruited from social care day centers and supported accommodation services across Northern Ireland.	19 interviews were conducted with 22 carers. 24 individual interviews with professionals. Five focus groups were held with 24 frontline staff. The information gathered from the group and individual interviews was analyzed using thematic content analysis.	Although there was agreement on the need for RSE, four barriers were commonly reported: the need to protect vulnerable persons; the lack of training; the scarcity of educational resources; and cultural prohibitions. The impact of these barriers could be lessened through partnership working across these groups involving the provision of training and information about RSE, the development of risk management procedures and the empowerment of people with intellectual disabilities.
14	[31] Lofgren-Martenson, L. Sweden	The aim of the article is to identify, describe and understand the opportunities and hindrances for young people with intellectual disabilities in forming relationships and expressing sexuality and love.	Study included 13 youngsters with ID, 13 staff members, and 11 parents, but also observation of dance gatherings.	Fourteen participant observations at dances geared towards youths with ID and qualitative interviews with youngsters, staff members and parents. Thematic and ethnographic analysis (cultural and historical context) using symbolic interactionism as a framework for analyzing observations in people’s dance venues. Dimension of analysis of the social construction of sexuality.	The results show a big variation of sexual conduct, where intercourse seems to be quite unusual. The study also shows that staff and parents feel responsibility for the youngsters’ sexuality and often act disciplinary as ‘new institutional walls’, while the youngsters develop different social strategies to cope with the surroundings. It seems clear that staff need more guidance and education about sexuality and disability in their social interaction with a new generation of people with ID.
15	[32] Maguire, K. et al. United Kingdom	This research aimed to explore support workers’ understanding of their role supporting the sexuality of adults with learning disabilities.	Six support workers from supported living services.	Six support workers were interviewed about their role. Data were analyzed using interpretative phenomenological analysis.	Support workers held conflicting beliefs and emotions about their role supporting sexuality. This was interpreted as creating an ambivalence that could result in support workers distancing themselves from an active role in supporting sexuality. Support workers may inadvertently express an understanding of their role that may be consistent with negative and limiting discourses about the sexuality of adults with learning disabilities.
16	[33] Pariseau-Legault, P. Canada	To explore the ethical implications of support workers’ experiences concerning sexuality in the context of intellectual disabilities in everyday practice.	Six support workers.	In-depth individual interviews analyzed thanks to critical phenomenology.	Support workers’ experiences related to sexuality in the context of intellectual disabilities are influenced by how they define their role in a clinical context. This role is influenced by how affective and sexual life is included in practices, local policies, and interdisciplinary work. Despite positive attitudinal changes, sexuality is still regarded as a sensitive topic capable of endangering both service users and support workers.
17	[34] Rohleder, P.; Swartz, L. United Kingdom	This article explores the challenges expressed by participants who provide sex education for persons with learning disabilities.	Four family planning social workers and three specialized teachers working at two different schools for learners with learning disabilities.	Thematic analysis of four in-depth interviews and of a group interview.	The findings reveal a tension between a human rights discourse and a discourse of restriction of sexual behaviors.
18	[35] Roy, M. et al. United Kingdom	This article examines the reasons for the gynecological consultation for sterilization of women with a learning disability (mental handicap).	Nine girls and women with a learning disability (mental handicap) and their caregivers.	Semi structured interviews with the girls and women with a learning disability. All professionals involved were interviewed individually. The interviews were analyzed using content analysis.	Most referrals were found to be initiated by the patient’s mother, where her own contraceptive history was found to be important in her assessment of the benefits of sterilization. It was felt that a comprehensive assessment and interviewing multiple informants enabled a more accurate assessment to be made about the woman’s ability to consent, and if she were unable to do so, then to assess her best interests. Most of the cases were found to be suitable for reversible noninvasive means of contraception.
19	[36] Santamaria, E. France	To question the recognition of the passage to adulthood of people with mental disabilities living in specialized institutions.	The director, psychologists, and educator of a medical-educational institute for people with mental disabilities in France.	Three-year observation of the daily life of the institution by participating in the activities of the professionals and the users, the times of meetings reserved for the employees or the meetings with the parents. Thirty one-hour semi-directive interviews with the director of the institution, psychologists, and some educators analyzed with thematic content analysis.	The topic of sexuality is a passionate one in institutions, involving a lot of difficulties. Educators must address sexuality issues to the same extent as they address knowledge of the body or the importance of nutrition. There are tensions and negotiations between the institution and the families (particularly with regard to contraception). It is important to break down the isolation of institutions and to offer training to professionals and build new projects together. Education and awareness work are necessary to change the way people with ID are perceived.
20	[37] Thompson, V.R. et al. Australia	This paper examines how the assessment tools clinicians are using have been developed to assess the sexual knowledge of people with ID.	Twenty-three clinicians using sexual knowledge assessment tools, working directly with people with ID in relation to their sexual health, some directly employed by disability service providers (*N* = 19) or in private practice (*N* = 4). A range of disability programs were represented, including residential (*N* = 6), day programs (*N* = 4), and behavioral support (*N* = 9).	Semi structured qualitative interviews analyzed with a constructivist grounded theory approach.	Assessment of sexual knowledge is not routine in disability service provision. Sexual knowledge is typically only assessed when there has been an incident of problematic sexualized behavior. This reactive approach perpetuates a pathological sexual health discourse. Clinicians also reported that the tools have gaps and are not fully meeting their needs or the needs of people with ID.
21	[38] Wilkenfeld, B.F. et al. United States	This study presents educators’ attitudes and beliefs towards the sexuality of adolescents and adults with developmental disabilities.	Five teachers in a school program and five instructors in an adult day services program at an educational facility for individuals with medically complex developmental disabilities.	Open-ended, structured interviews analyzed with content analysis coding.	Results indicate that educators hold a positive view towards providing sexuality education and access to sexual expression for persons with developmental disabilities. Educators viewed sexuality as a basic human right, yet expressed concerns regarding capacity to consent to and facilitation of sexual activity. This study helps to understand barriers preventing the delivery of sexuality education to this underserved population.
22	[39] Yool, L. et al. United Kingdom	The attitudes of staff toward the sexuality of adults with learning disabilities within a medium-secure hospital within the United Kingdom were examined using qualitative research methods.	Four full-time staff members: consulting psychiatrist, long-term care worker, advocacy worker and domestic staff member.	Interviews were transcribed and analyzed using a method adapted from phenomenology.	The analysis revealed that staff members generally held liberal attitudes with respect to sexuality and masturbation. However, with respect to sexual intercourse, homosexual relationships, and the involvement of adults with learning disabilities in decisions regarding their own sexuality, a less liberal attitude was detected. Concern was also noted with respect to the attitudes of female staff members towards the sexuality of adults with learning disabilities who have committed sexual offenses. Training issues were also identified and implications for the service were discussed.
23	[40] Young, R. et al. United Kingdom	To examine whether gender of people with ID affects staff attitudes regarding the sexuality of people with ID.	Ten staff members of specialized services for people with ID.	Semi structured interviews were completed with 10 staff members and analyzed using thematic analysis.	The study indicates unfavorable attitudes towards sexuality in individuals with ID that correlate with traditional, restricted gender stereotypes. The identification of these themes highlights the importance of considering gender when supporting the sexuality of people with ID.
24	[41] Szollos, A.A.; McCabe, M.P. Australia	This study compares the sexual and relational knowledge, experience, feelings and needs of people with ID with the perception of their caregivers and the knowledge and experience of people without ID.	The sexuality of 25 individuals with mild ID was assessed based on interviews that addressed their knowledge, experience, feelings, and needs. Compared to 39 college students and 10 care staff.	Data on the sexuality of a group of people with mild ID were obtained by interviewing them directly thanks to the Measure to Assess Sexual and Relational Knowledge, Experience, Feelings and Needs (Sex Ken-ID), and then comparing their responses with the perceptions of their caregivers, as well as to data collected from a group of people without ID.	Care staff consistently overestimated the responses of their clients, whom they perceived to be more knowledgeable and experienced, have more positive feelings, and a greater need to know than was indicated by the clients themselves. The group without ID demonstrated a higher level of sex knowledge and reported greater interactive sexual experience than the people with ID. The exceptions to this were that the latter group had experienced higher levels of sexual abuse, and reported equal frequencies of same-sex experiences.

## Data Availability

Available upon request from the corresponding author.

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
