# Peer review of "Representations of Sexuality among Persons with Intellectual Disability, as Perceived by Professionals in Specialized Institutions: A Systematic Review"

_ijerph, 2022, doi:10.3390/ijerph19084771_

Round 1

Reviewer 1 Report

I would like to congratulate the authors for focusing on this very complex and sensitive subject for their research and this article. The article is well structured and provides a good overview of the research process. I would suggest some of the following aspects to consider to make this a stronger article.

There is clear information of the search criteria of the literature and the reasons for excluded some of the sources at different stages. It would be beneficial if an inclusion and exclusion criteria was provided. For example, it was not clear until towards the end of the article that year of publication was not considered as an exclusion criteria. Having clear information about the criteria will help readers to understand the process of selection better.

There are some challenges in writing for an international audience, especially when it comes to terminology. Clarification on how learning disabilities is described would be useful. I am assuming this refers to dyslexia, but since the same term is used for intellectual disabilities in the UK, clarifying this is necessary. (Page 5- lines 183-4).

In terms of presentation of the results, systematic reviews tend to present a table with all the sources and the key information about each article reviewed. Lack of such a table makes it harder for a reader to understand which articles every theme was extracted from. I would recommend including a table of all the literature with the key information.

The last section of the third theme (3.2.3) seems to change its focus to training issues specifically. I wonder whether this needs to be presented as a distinct sub-theme or theme rather than embedding this discussion into the current theme. Doing this would also help in making some the points raised in the discussion more clearly linked to this theme.

There is some repetition of points and information which can be removed with more careful proof-reading.

Author Response

Thank you very much for your careful reading of our manuscript and the precious reviews you gave. 

We clarified the inclusion and exclusion criterion in the method section. Doing so, we explained more carefully the diagnosis of intellectual disability in order to help the readers understand the co-occurence of this mental retardation with other mental conditions. We defined specifically learning disabilities and explained why we included articles refering to this mental condition in the systematic review. 

We added a table with all the sources and key information as you suggested.

In the thematic analysis, we chose to keep the sub-theme on the training as it was initially presented. However, we added a few lines on this sub-theme in the discussion section. 

Finally, we proofread and modified the article to remove repetitions. 

Best regards, 

Reviewer 2 Report

I enjoyed reading this systematic review of literature relating to sexuality and intellectual disability. The paper reflects the literature well, and is founded on an appropriately systematic methodology, which is, for the most part, well described. 

The discussion of search terms is very general, considering the very specific focus of the paper - whilst it is made clear that this is therefore one part of a wider study, this more general discussion takes up a significant amount of space.  In respect of the specifics of the filtering for this paper, on p. 5 (lines 181-193) the authors say they excluded 48 (of 96) papers, but then describe rejecting 40 as out of scope, 10 due to methodological weakness, and 1 duplicate - which totals 51 exclusions - this needs to be checked. 

The authors make clear that they didn't want to place artificial date boundaries around the systematic review, but it would nevertheless be useful for the reader to know more about the timeframes of the selected articles. What is the date range of these publications? Are they evenly spread through that date range? Are there particular temporal clusters of articles? 

On p. 9, the concept of a 'system of incompatibility' is introduced - it would help the reader to know earlier in the analysis, and in the abstract, that this framework is considered helpful in explaining the paradox. It would also be useful for English speaking readers to have a somewhat enlarged description of this concept.

At lines 199-208 quantitative studies were excluded, but these were then described as corroborating the qualitative findings at lines 363-370, which suggests that the quantitative studies were also read and analysed. It would, therefore, be useful to have some systematic analysis of those quantitative studies.

At lines 430 - 432 a claim is made regarding the effectiveness of training that doesn't quite come through in the analysis above. Some evidence for this claim would be helpful.

Overall, this is a useful systematic review of the literature, that with some tidying up and strengthening of the analysis will make a useful contribution to the literature.

Author Response

Thank you very much for your careful reading of our manuscript and the precious reviews you gave. 

First of all, we have corrected the number errors on page 5.

Secondly, we clarified the inclusion and exclusion criterion in the method section. We thus clarified the time frame of the articles we included.

About the 'system of incompatibility', we introduced this notion in the abstract. But we decided not to enlarge the description of this notion in this systematic review. It is precisely the analysis of the literature that led us to use this concept, which seemed to us relevant to better understand the phenomenon. Introducing it earlier in the analysis would be more interesting but would go somewhat against the way the systematic review operates. We count on deepening this notion in a future article where we discuss our own research material.  

Both in the method section and in the discussion section, we rephrased to clarify the use of the quantiative studies. They were excluded of the systematic review, but we referred to some of them in the discussion to deepen the analysis. 

And finally, we added a few lines in the discussion on the question of training.

Thanks again for you reviews. Best regards,  

Reviewer 3 Report

The authors have written an interesting paper that is certainly worthy of publication.

There are two clarifications necessary in the methodology:

At line 192: what methodological weaknesses are these?

“This article will focus on the 22 qualitative studies because they allow a better understanding of the subjective experience of professionals. We believe that this is a central aspect of a rather complex issue. We also incorporated the two mixed methods studies and were particularly interested in their qualitative aspects.” – I don’t understand: if you were never going to include quantitative studies, why not have that as an exclusion criterion? The explanation you provide here is really confusing. If you only include qualitative studies, that should be part of your study goal, no? But nowhere before in the text is there any indication that you will only include qualitative studies.

Author Response

Thank you very much for your careful reading of our manuscript and the precious reviews you gave. 

Firstly, at line 195, we gave more details on the reasons why articles have been excluded of the systematic review for methodological weakness.

Secondly, both in the method section and in the discussion section, we rephrased and clarified the use of the quantitative studies. They were excluded of the systematic review, but we referred to some of them in the discussion to deepen the analysis. 

Thanks again for you reviews. Best regards,